# Deciphering genetic susceptibility to clear cell renal cell carcinoma

Maria Mandelia ⓘ, Philip J. Law ⓘ, Charlie Mills ⓘ, Molly Went ⓘ, Jayaram Vijayakrishnan & Richard S. Houlston ⓘ ✉

Genome-wide association studies (GWAS) have identified over 60 autosomal risk loci associated with clear cell renal cell carcinoma (ccRCC), yet the functional mechanisms underlying these associations remain largely unclear. To establish connections between risk variants and their target genes, we applied the activity-by-contact (ABC) model, which integrates epigenomic data and Micro-C interactions, complemented with renal-specific quantitative trait loci, to predict enhancer-gene relationships. Our analyses implicate variation in hypoxia sensing, cell cycle regulation, and telomerase maintenance pathways as central mediators of ccRCC risk. These findings provide new insights into the molecular basis of ccRCC susceptibility and highlight potential therapeutic avenues for prevention and treatment.

Kidney cancer (renal cell carcinoma, RCC) is a common malignancy with an estimated 430,000 new cases diagnosed worldwide in 2020[1]. Over 70% of RCCs are clear cell RCC (ccRCC) tumours characterised by loss of function of the von Hippel-Lindau (VHL) tumour suppressor with consequent unrestrained activation of hypoxia-inducible transcription factors (HIF), which in turn leads to the over-expression of genes that drive cell growth[2,3].

Rare mutations in several genes confer high penetrance susceptibility to ccRCC (including *VHL, BAP1, FLCN, PTEN, MET, SDHD, ELOC, TSC1*)[4–6], however these account for <5% of cases and little of the two-fold familial risk. Our understanding of the heritable basis of RCC has recently been transformed by genome-wide association studies (GWAS), demonstrating the existence of common low penetrance susceptibility[6,7].

A major aim of cancer GWAS is to identify genes and functional mechanisms that may ultimately provide clinically useful targets, for example in chemoprevention. Functional characterisation of selected RCC risk loci have provided evidence for local biological mechanisms underlying specific associations, including allele-specific expression of *CCND1* (11q13.3)[7,8], *BHLHE41* (12p12.1)[9], and *DPF3* (14q24)[10], as well as differential binding of HIF transcription factors and PAX8 in an allele-dependent manner[11–13]. Determining the genetic and functional basis of GWAS risk loci for cancer has, however, been challenging due to linkage disequilibrium between variants and the localisation of most risk variants to noncoding regions. These causal variants are often regulatory, affecting the expression of multiple target genes. Although tissue-specific expression quantitative trait locus (eQTL) data has aided in the identification of target genes, existing eQTLs only capture a small fraction of the GWAS heritability of cancers. To the extent that GWAS risk loci have been elucidated, incorporation of functional biology data has been demonstrated to improve the identification of causal variants and respective target genes. The activity-by-contact (ABC) model[14,15] has computationally formalised this approach by linking regulatory elements to target genes through enhancer activity and 3D chromatin contact frequencies.

To better understand ccRCC risk associated with common genetic variation, GWAS data were integrated with multiple data modalities, utilising the ABC model. Our analysis directly links genetic risk variants to target genes, highlighting potential therapeutic targets for prevention and treatment (Fig. 1).

## Results

### GWAS statistics and definition of risk loci

We reanalysed data from a recent GWAS meta-analysis of renal cell carcinoma (RCC)[6], restricting the data to the clear cell RCC (ccRCC) from European ancestry individuals (14,627 cases and 738,190 controls). To identify secondary association signals, we performed a conditional analysis. We identified 52 independent risk loci ($P < 5 \times 10^{-8}$) localising to 46 regions of the genome. These included 48 risk loci (44 regions) reported by Purdue et al., but we excluded 30 risk loci (8 regions) that were shown to be non-independent on re-analysis (i.e. $r^2 > 0.01$) or non-significant in European ancestry individuals (Supplementary Data 1). We additionally recovered novel associations at 3p11.1 and 5q35.1, which were not previously reported but are independent ccRCC risk loci in individuals of European ancestry, as well as novel conditional variants at 7p22.3 and 22q12.1.

### Cell-specificity and chromatin landscape of risk loci

To gain insight into the cellular context of the risk loci, we analysed single-cell RNA sequencing (scRNA-seq) profiles across 24 different tissue types

Division of Genetics and Epidemiology, The Institute of Cancer Research, Sutton, UK. ✉e-mail: richard.houlston@icr.ac.uk

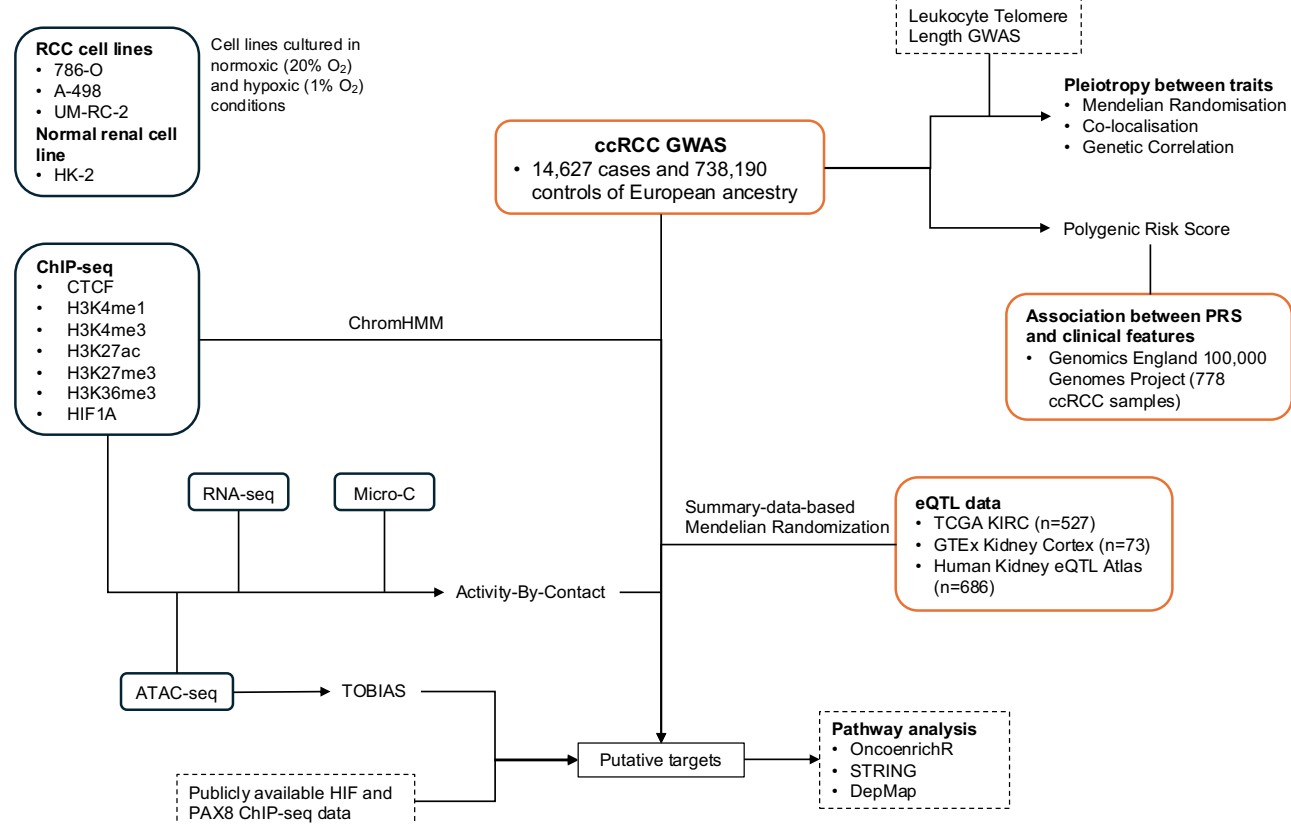

**Fig. 1 | Overview of the study.** Using published clear cell renal cell carcinoma (ccRCC) GWAS data, we identified 52 regions of interest. Data from epigenetic marks (ChIP-seq), chromatin accessibility (ATAC-seq), gene expression (RNA-seq) and long-range chromatin interactions (Micro-C) were combined using the Activity-by-Contact (ABC) model to identify putative genes associated with the risk loci. These variants were linked to target genes by analysing renal-specific eQTLs and using Summary-data-based Mendelian Randomization (SMR). Boxes with solid lines indicate data generated in-house in renal cell lines, boxes with red lines indicate data from patient data, and boxes with dashed lines indicate publicly available data resources.

using the Tabula Sapiens dataset[16]. We derived single-cell disease relevance scores (scDRS), which link the scRNA-seq data with polygenic risk at single-cell resolution[17]. Genes whose expression was correlated with scDRS were enriched in kidney and renal epithelial tissue ($P_{adj} < 0.05$, Supplementary Fig. 1). GWAS risk variants typically influence tissue-specific gene expression through *cis*-regulatory mechanisms. We therefore sought evidence of enrichment of histone H3 lysine 4 trimethylation (H3K4me3), H3 lysine 4 monomethylation (H3K4me1) and H3 lysine 27 acetylation (H3K27ac) marks using ChIP-seq in renal cancer cells as well as chromatin data in 127 cell types from the Roadmap Epigenomics Project[18]. Collectively, analysis of these data confirmed significant enrichment of risk loci specific to renal tissues in enhancer and promoter-associated histone marks ($P < 10^{-4}$; Supplementary Fig. 1).

**Functional fine-mapping of risk loci**

To prioritise causal variants at each ccRCC risk association we fine-mapped each of the risk loci, incorporating functional information using PolyFun[19] and susieR[20]. Credible sets of causal variants were identified by susieR using the priors calculated by PolyFun. Posterior inclusion probabilities (PIP) were ranked, and variants added to the set until the cumulative PIP > 0.95, with a minimum individual variant PIP of 0.001. This analysis resolved nine risk loci, with each locus containing 1-5 credible sets (median = 1), and each set consisting of 1-44 variants (median = 1). (Supplementary Data 2). Among the analysed variants, rs10828248 (10p12.31), rs4903064 (14q24.2), rs4389139 (16q12.1), and rs2860183 (19p13.2) showed direct evidence of regulatory effects, supported by data from a contemporaneous massive parallel reporter assay (MPRA) pan-cancer study[21] (Supplementary Data 3).

To further refine loci that contained correlated variants, we annotated variants with transcription-factor (TF) binding information, focusing on pathways relevant to ccRCC biology, particularly the hypoxia-inducible factor (HIF) pathways, which is dysregulated in ~80% of ccRCC cases due to VHL loss (Supplementary Data 1)[2]. Using the JASPAR 2024 TF binding database[22] we surveyed chromatin accessible regions and identified HIF-related TF binding sites (HIF1A, ARNT, also known as HIF1B, and EPAS1, also known as HIF2A) at the 12p12.1 and 22q13.31 loci. These predictions were validated using publicly available and in-house ChIP-seq data (Fig. 2; Supplementary Data 1). Notably, at the 11q13.3 locus, strong HIF1A and EPAS1 ChIP-seq peaks overlapped the risk variants in the RCC cell line 786-O under hypoxic conditions (1% $O_2$), but not in MCF-7 breast cancer cells[8] or normal renal HK-2 cells under normoxic conditions (20% $O_2$), consistent with prior reports (Fig. 2). EPAS1 binding was also observed at 7p12, 8q24.21, 11q13.3 and 12p12.1 risk loci, which has been documented to be preferentially recruited to PAX8-bound transcriptional enhancers[11]. PAX8 binding was identified at the 11q13.3[11], 8q24.21 and 10p12.31 risk loci (Supplementary Data 1).

To evaluate allele-specific effects on HIF binding motifs we used motifbreakR[23]. Among the risk loci predicted to harbour HIF1A motifs (12p12.1 and 22q13.31), only the variant at 22q13.31 (rs714024) was predicted to have significant allele-specific difference in binding affinity ($P = 6.81 \times 10^{-5}$). Analysis of HIF ChIP-seq data further indicated that the 22q13.31 locus was also bound by ARNT and rs714024 was predicted to disrupt ARNT binding ($P = 6.7 \times 10^{-5}$), as was rs6442146 at 3p25.3 ($P = 4.6 \times 10^{-5}$). At loci overlapping HIF1A ChIP-seq peaks, rs6442146 (3p25.3, $P = 1.0 \times 10^{-3}$) and rs11643164 (16q12.1, $P = 3.6 \times 10^{-5}$) were also

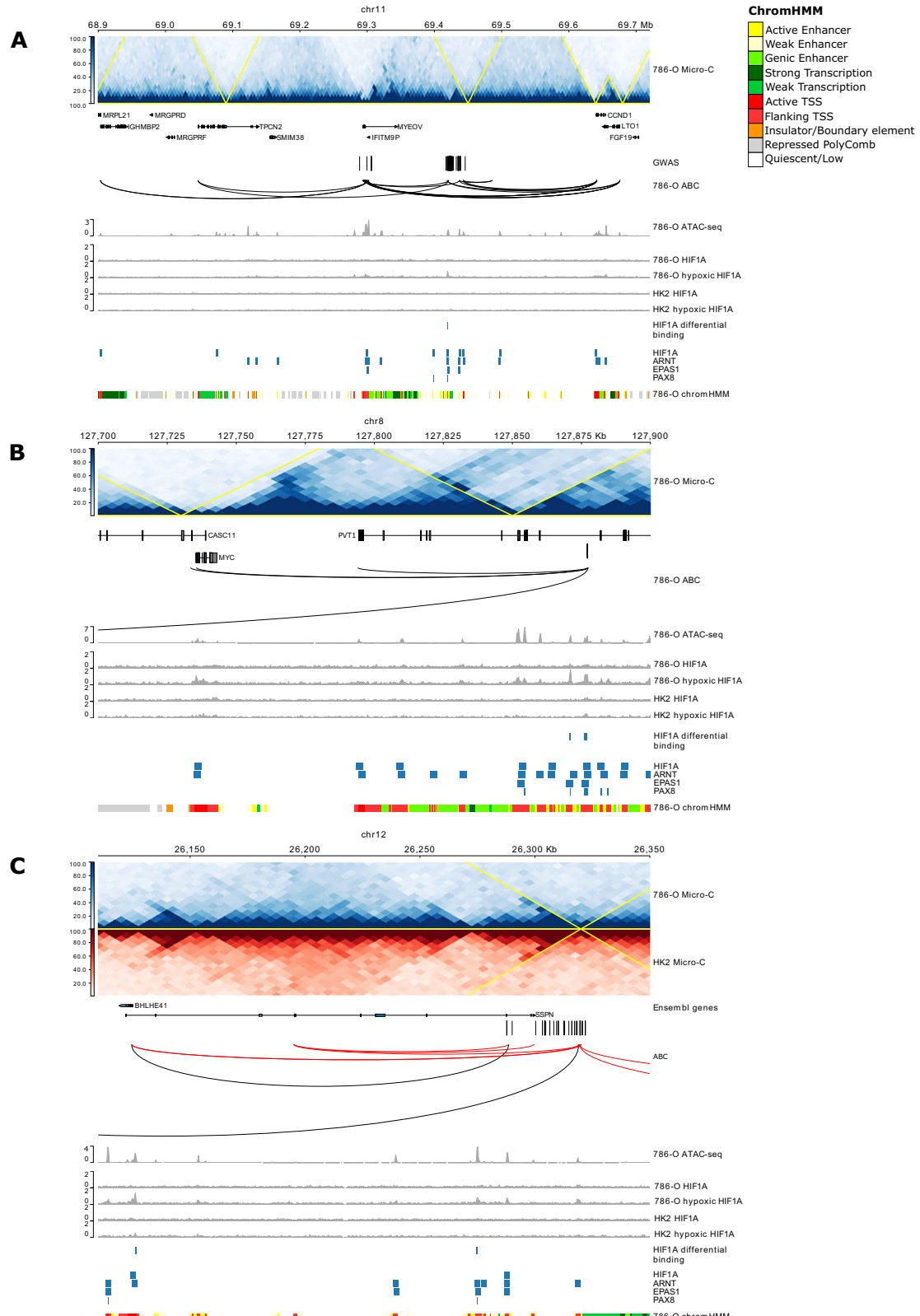

**Fig. 2 | Regional plots showing annotation of (A) 11q13 (*CCND1*), (B) 8q24 (*MYC/PVT1*), and (C) 12p12 (*BHLHE41*) risk loci.** For each sub-figure from the top are the Micro-C contact map with predicted topological associated domains (TAD) overlaid in yellow, gene models, GWAS variants ($r^2 > 0.8$ to lead variant), Micro-C chromatin loops, ATAC-seq peaks, HIF1A peaks in normoxic and hypoxic conditions for 786-O (renal carcinoma) and HK-2 (normal) cell lines, regions of differential binding between HK-2 and 786-O, HIF1A, ARNT (also known as HIF1B), EPAS1 (also known as HIF2A), and PAX8 peaks from publicly available data, and the predicted chromHMM states. In addition, for sub-figure C, the Micro-C contact map and loops for HK-2 are shown in red. For clarity, only protein-coding genes are shown. Coordinates are in GRCh38.

predicted to disrupt binding. No EPAS1 ChIP-seq peaks overlapped with risk variants showing significant allele-specific effects on EPAS1 binding.

To quantify differential TF binding under normoxic and hypoxic conditions, we performed a comparative analysis of HIF1A ChIP-seq data in 786-O and HK-2 cells. Of the 2,692 HIF1A binding sites 42.5% (1153 sites) were differentially bound under hypoxic conditions (FDR < 0.05), with three sites overlapping risk loci at 3q11.1, 7p22.3, and 11q13.3. By contrast, only 8.7% of the peaks were differentially bound under normoxic conditions, none of which overlapped GWAS loci. Motif enrichment analysis of binding sites revealed several candidate cofactors, among which SP1, was notable. SP1, a TF implicated in angiogenesis, proliferation[24], telomere maintenance[25], and AKT-MYC signalling[26], was most frequently predicted, with binding sites at 2q21, 10p12.31, and 16q12.1.

## Identification of target genes

After identifying likely causal variants and cell types, we prioritised target genes. We initially focused on risk variants within coding regions, and hence those likely to have a direct effect on the expressed protein a priori. The missense variants rs2277283 (M > T, INCENP, 11q12.3) and rs116483731 (R > Q, SPDL1 (also known as CCDC99), 5q35.1) were identified as likely causal. CADD predicted rs2277283 as pathogenic (CADD score: 23.5), supported by AlphaMissense[27]. As INCENP is an inner centromere protein[28], this change suggests potential disruption of the metaphase-anaphase transition. rs116483731 (CADD score: 24.2) was predicted to be deleterious by CADD but benign by AlphaMissense, yet its role in mitotic spindle formation and chromosome segregation[29], combined with rare INCENP germline mutations previously being linked to juvenile nephronophthisis[30], underscore their relevance to renal biology.

We linked an additional 17 risk loci using supporting evidence from expression analysis based on eQTL data. We performed a Summary-data-based Mendelian Randomization analysis (SMR)[31] using GTEx Kidney cortex (73 healthy individuals)[32], Susztak Lab Human Kidney eQTL Atlas (686 non-neoplastic kidney tissues)[33], and TCGA KIRC (kidney renal cell carcinoma, 527 individuals)[34] identifying candidate target genes for the risk loci ($P_{SMR}$ < 0.05) across all the datasets (12 from the kidney tissue meta-analysis, seven from the TCGA KIRC samples, and three from GTEx; Supplementary Data 1).

To link variants at each of the other risk loci to respective target susceptibility genes we evaluated the quantitative effect of enhancer-gene regulation in 786-O cells by analysing Micro-C data in conjunction with ATAC-seq, H3K27ac ChIP-seq and RNA-seq data using the ABC[14,15] model. 25 loci were predicted to regulate a gene, with these interactions regulating 4 genes on average (median = 2 genes), with a mean distance of 186 kb (median distance = 71 kb), and these gene-enhancer predictions were largely consistent (15/25 loci) across RCC (786-O, A-498, and UM-RC-2), and normal kidney (HK-2) cell lines, consistent with shared chromatin architecture (Supplementary Data 4). We compared our ABC model predictions with published enhancer-gene pairs from 811 tissues or cell lines in ENCODE[35]. While we did not observe substantial tissue specificity, there was variant specificity. For example, at the 16q12.1 locus (rs11643164), the enhancer region association with HEATR3 was found across all cell types. In contrast, the 8q24.21 locus (rs6470589) association with PVT1, CASC11, and MYC was only found in RCC and renal tissues (Supplementary Fig. 2). These data imply that the chromatin interactions underlying RCC risk are a fundamental feature of renal cell biology.

We prioritised the gene mapping evidence in the following order: coding variants, SMR, ABC, and finally the closest gene. These categories were generally mutually exclusive. For instance, missense variants generally lacked SMR associations or ABC predictions. Similarly, several genomic loci with an SMR result did not demonstrate a confident enhancer-gene link. For example, the 1p36.21 locus was exclusively associated with CASP9 by SMR. When multiple gene predictions were available, the consensus prediction was selected. For example, at the 22q13.31 locus, where SMR indicated GRAMD4 and ABC implicated both CERK and GRAMD4, GRAMD4 was chosen. Where discordant predictions occurred, the union of all putative

genes was retained. Based on these data combined with the genes containing coding variants we identified high confidence target genes for 29 of the 52 ccRCC risk loci. For the remaining loci, we assigned the closest gene as lower confidence gene predictions, 13 loci of which fell within introns of genes. In total, we identified 61 target genes as the plausible functional basis of the ccRCC risk loci. Finally, we examined whether any of the target genes overlap with genes that contain recurrent somatic ccRCC driver mutations[4,5], only finding TERT.

Using this approach, we validated MYC-PVT1, CCND1, BHLHE41 and DPF3 as the biological basis of the previously reported 8q24.21, 11q13.3, 12p12.1 and 14q24 associations, respectively (Fig. 2; Supplementary Fig. 3)[2,8–11]. In addition, at the 11q13.3 locus, we identify a conditional GWAS signal that is associated with MYEOV, a known cancer related gene[36,37] (Fig. 2). Our analysis provided evidence to implicate multiple genes as determinants of ccRCC risk. Germline coding variants in CHEK2, a cell cycle checkpoint regulator, play well-established roles in susceptibility to multiple cancers, including breast[38], prostate[39], and pancreatic cancers[40]. The ABC loop links the risk locus rs9625647 (22q12.1), which is intronic to ZNRF3, with the TSS of CHEK2, and we identified reduced expression of CHEK2 as compared to normal renal cells (logFC = -0.42, $P_{adj}$ = 3.37 × 10$^{-3}$; Supplementary Data 5). Since ABC did not predict this loop in the normal HK-2 cell line, this suggests that this interaction may be an acquired feature associated with tumour development (Supplementary Fig. 3). Further examples include the chromatin interactions implicating the transcription factor MLLT10 as the basis of the rs7084454 (10p12.31) association. Although not previously linked in the development of ccRCC, variation in MLLT10 has been reported to be associated with risk of meningioma[41]. Similarly, ABC links rs11643164 (16q12.1) to HEATR3, a key regulator of ribosome biogenesis, as a candidate target gene for ccRCC susceptibility, with a prior evidence for an association with glioma risk[42,43] (Supplementary Fig. 3).

Overall, these candidate target genes exhibited a stronger set of functional interactions than expected by chance (STRING database[44]; 44 versus 24 expected interactions, $P$ = 1.87 × 10$^{-4}$). An analysis of bulk RNA-seq of RCC cells showed 29 were differentially expressed compared to normal ($P_{adj}$ < 0.05; Supplementary Data 5). The most-enriched biological pathways included those related to cell cycle control and signal transduction (Supplementary Data 6). After investigating regulatory networks in the gene list, we identified sets of 'hub' genes including MYC-CCND1-CDK6, TERT, as well as those related to transcription factors involved in the induction of oxygen regulation (EPAS1) or cell cycle (SP1, CDKN1A) (Supplementary Fig. 4).

## Longer telomeres mediate risk

Amplification or epigenetic modification of TERT (rs7734992, 5p15.33) plays a crucial role in ccRCC by maintaining cellular immortality, with HIF1A and EPAS1 regulating TERT expression, enhancing tumour survival under hypoxic conditions[45]. Hence it is intriguing that as well as TERT, another of the target genes, TERC (rs55735727, 3q26.2), has a well-established role in telomere maintenance. As leucocyte telomere length (LTL) has an established role in genome stability[46], we examined for evidence of pleiotropy between ccRCC and LTL by analysing data on 472,174 individuals from the UK BioBank[47] using two-sample Mendelian Randomisation (MR)[48,49]. Increased LTL was consistently associated with increased ccRCC risk (inverse variance weighted random effects $P$-value = 2.48 × 10$^{-17}$; Fig. 3; Supplementary Data 7), with the Steiger test confirming that this was the most likely causal direction. We performed a colocalisation analysis[50,51], finding evidence of shared causal variants for increased ccRCC risk and increased LTL at the TERT and TERC loci (Supplementary Data 8). In addition to these two associations, an additional six LTL loci were associated with ccRCC at region-wide significance (i.e. $P$ < 3.45 × 10$^{-4}$; 0.05/145 LTL loci), including telomere related genes RTEL1 and POT1. To examine whether this apparent genetic pleiotropy was reflected genome-wide, we performed a genetic correlation using LDAK[52]. Although the estimated genetic correlation between ccRCC and LTL using

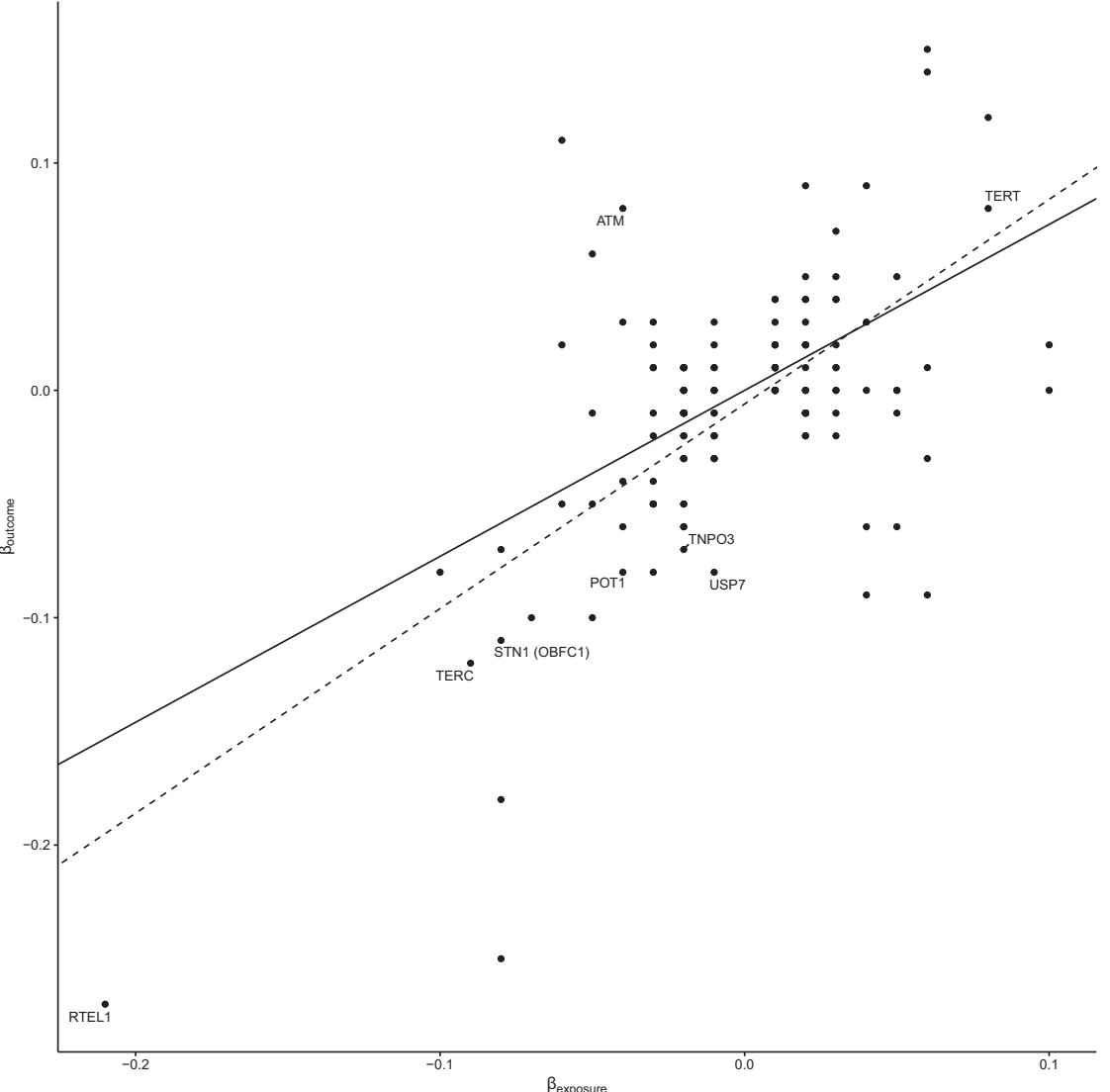

**Fig. 3 | Pleiotropy with lymphocyte telomere length (LTL).** Mendelian randomisation (MR) plot showing effect sizes ($\beta$) of LTL in the UK Biobank (exposures) and their effect sizes ($\beta$) on ccRCC risk (outcome). Lines correspond to the slopes of the tests: inverse-variance weighted (solid), and MR-Egger (dashed).

LDAK was not strong ($R_g = 0.05$) and did not reach statistical significance, it suggests a positive trend. Taken together, these findings support the notion that a subset of genetic variants may increase ccRCC risk by promoting longer telomere length, potentially enhancing replicative lifespan or chromosomal stability, and thereby facilitating neoplastic transformation.

### Impact of risk variants on somatic alterations and clinical features

The impact of rare high-penetrance germline susceptibility variants on tumour development is often reflected in the somatic mutational profile of tumours (e.g. colorectal and breast cancer) and clinical phenotype[53,54]. To explore the possibility of a similar relationship with ccRCC risk variants we analysed 778 ccRCC patients recruited to the 100,000 Genomes Project[55] (100kGP) using previously generated data[5]. For each patient we generated a polygenic risk score (PRS) based on the ccRCC GWAS summary statistics, and examined the relationship between PRS with clinical features (age at sampling, tumour grade and stage), somatic alterations (ploidy, telomere length and telomeric-to-non-telomeric content ratio) as well as single base substitution (SBS) and double-base-substitution (DBS) and indel (ID) mutational signatures. Analyses of these data provided no support for PRS influencing any of these features (Supplementary Data 9). Sample numbers

however precluded examining for a possible relationship between PRS and presence of a specific driver gene mutation.

### Gene list analysis

To determine which of these candidate target genes we identified have an established role in oncogenesis, and more specifically in ccRCC, we used the text mining tool OncoScore[56], which examines text from all available studies in the biomedical literature (Supplementary Data 10). To complement this search, we queried semantic predications within the Semantic MEDLINE Database[57] using MELODI Presto[58]. Integration of these data revealed that, while many of the genes have previously been implicated in cancer, the majority (44/61) of the candidate target genes presently have no documented role in ccRCC (Supplementary Data 11).

We interrogated these candidate genes in the Genomics England ccRCC dataset to evaluate the impact on patient outcome. Only seven genes had pathogenic or likely pathogenic germline mutations, namely *TERT* (12% of patients with this mutation), *CHEK2* (3%), *MAD1L1* (2%), *INSR* (1%), *PROS1* (0.4%), *LRP2* (0.3%), and *KCNQ1* (0.1%). Of those that were mutated in >10 patients (*TERT, CHEK2, MAD1L1, INSR*), we performed a univariate linear regression for grade and stage, and none showed a significant association. Similarly, we performed a Cox proportional-hazards

model test (with all passing the model assumptions), adjusting for age and sex, and none were significantly associated with overall patient survival.

One of the aspirations of GWAS is to inform therapeutics. To investigate the potential clinical utility of the ccRCC target genes we implicated, we used oncoEnrichR[59] to explore multiple sources of functional and drug curation, including Open Targets[60], DrugBank and DepMap[61]. Based on CRISPR knockout data, genomic biomarkers and patient data, several target genes were identified as attractive drug targets, including *CCND1, INCENP*, and *EPAS1* represent potentially promising therapeutic targets in ccRCC (Supplementary Data 12). For several genes, there are already approved drugs that provide an opportunity for repurposing, including Palbociclib, metformin, mefloquine, and quinidine. Palbociclib is currently approved for use in breast cancer and targets CCND1 through CDK4/6 inhibition[62]. The diabetic therapy, metformin directly targets INSR, which is a component of the insulin signalling pathway[63]. The antimalarial mefloquine and the antiarrhythmic quinidine both inhibit the potassium channel protein KCNQ1[64]. In addition to these there is evidence that EPAS1 may be indirectly targetable by the antifungal agent ketoconazole[65].

Using eQTL and MPRA data, we assessed the direction of gene expression for druggable targets identified via oncoEnrichR. The eQTL analysis from TCGA KIRC ($P_{adj} = 7.87 \times 10^{-7}$), GTEx Kidney cortex ($P_{adj} = 1.31 \times 10^{-6}$) and Susztak Lab Human Kidney eQTL Atlas ($P_{adj} = 3.96 \times 10^{-11}$) datasets showed that the risk allele of rs140527149 (1p36.21) is associated with reduced *CASP9* expression (Supplementary Data 12), an apoptosis related gene, suggesting that agonists to restore function could be beneficial. Similarly, MPRA data indicates that rs4389139 ($r^2 = 0.88$ to candidate variant, rs11643164, 16q12.1) affects *HEATR3* expression, and rs2860183 ($r^2 = 0.99$ to candidate variant, rs11085214, 19p13.2) affects *INSR* expression in RCC (Supplementary Data 3). An MPRA association supported a relationship between rs77247065 and *CCND1* expression but was not formally significant after multiple testing correction ($P_{adj} = 0.08$). Additionally in the RNA-seq data, 16 of the 25 potentially druggable genes, including *CDKN1A, TERT, CCND1*, and *INSR* were differentially expressed in RCC compared to normal kidney cells, indicating suitability for inhibitors or agonists (Supplementary Data 5).

Given the central role of these highlighted pathways in ccRCC development, these findings expand opportunities for therapeutic targeting or modulation, for example Imetelstat, an indirect telomerase inhibitor is in trials for myelofibrosis[66]; HSP90 inhibitors (e.g. ganetespib[67]) and EPAS1 antagonists. In addition, several natural compounds have shown some promise as potential adjuvants or chemotherapeutic agents (e.g. cumcumim, genistein, silibinin[68]) through indirect targeting of these pathways. Hence, adapted forms or modified dosing regimens of these drugs may offer alternative treatment options.

## Discussion

Herein we provide evidence to implicate 61 genes as the functional basis of ccRCC risk loci. For over a half of the GWAS risk loci, the candidate target gene was the closest to the gene or intronic, aligning with the Open Targets gold standard dataset[60]. This proximity effect has previously been noted and proposed to reflect evolutionary conservation[69].

We focused on European ancestry GWAS data due to the larger sample size (14,627 cases, 738,190 controls), which provided greater statistical power for fine-mapping and target gene identification. The original GWAS meta-analysis[6] included non-European populations (namely, Asian, Latin American, and African ancestries), but these had much smaller sample sizes, therefore limiting the detection of independent signals. We performed a preliminary analysis of the 52 identified loci using summary statistics from Asian (621 cases, 86,796 controls), Latin American (1277 cases, 2180 controls), and African (417 cases, 3109 controls) ancestry groups from Purdue et al. (Supplementary Data 13). Of the 52 loci, 6 showed nominal significance ($P < 0.05$) in Asian populations, 14 in Latin American populations, and 7 in African populations, with effect sizes generally consistent but weaker due to lower power. Notably, rs10908176 (*CCND1*, 11q13.3) and rs11085214 (*INSR*, 19p13.2) showed associations

across multiple ancestries, suggesting shared tumourigenic aetiologies. However, loci like rs116483731 (*SPDL1*, 5q35.1) and rs143459581 (*TTC28*, 22q12.1) were specific to European ancestry, potentially reflecting population-specific regulatory variants. Conversely, as previously observed[6], rs7629500 (3p25.3) is predominantly found in African ancestry populations (risk allele frequency = 0.1, <0.001 in European ancestry populations), and localises within the 3'-UTR of *VHL*, a known driver of ccRCC development.

Our analysis provided evidence to implicate multiple genes as determinants of ccRCC risk. Although many of the risk loci have not previously been the subject of detailed scrutiny, several of these target genes have either well documented roles in ccRCC or are strong candidates for having a role in tumour biology. For example, we identify rs1125068 as the causal variant underlying the 2p21 association, where the enhancer containing this risk allele interacts with the transcription start site (TSS) of *EPAS1* (also known as *HIF2A*). For several risk loci, there were no obvious candidate genes, primarily due to a paucity of functional data. This may also be indicative of alternative mechanisms of action underlying associations that we did not explore. For example, it has recently been proposed that a variable number tandem repeat within intron 6 of *TERT* altering splicing is a mechanistic basis of the 5p15.33 association[70].

The shared enhancer-gene connections shown in RCC and normal renal cell lines suggest that risk variants modulate pre-existing regulatory architecture rather than acting as primary oncogenic drivers per se. These variants are therefore likely to fine-tune gene expression (e.g. by altered TF binding or enhancer activity) in pathways such as hypoxia sensing (EPAS1) or cell cycle (CCND1), predisposing cells to tumourigenesis following additional somatic alterations (e.g. VHL loss). For instance, at 11q13.3, the risk variant rs10908176 is linked to enhanced *CCND1* expression via EPAS1 binding[8,11], which is a feature in both normal and tumour cells but amplified in RCC under hypoxic conditions. Such a model of predisposition is supported by the absence of recurrent somatic driver mutations in most of the target genes (except for *TERT*), consistent with their role in early susceptibility rather than direct transformation.

We recognise limitations in our study. Notably for many of the loci, we did not have eQTL data to support candidacy of the target gene. While we have sought to address the cellular context of eQTLs, analysing both normal and tumour data in kidney, a failure to demonstrate a relationship may simply reflect a lack of statistical power. Therefore, rather than rely solely on eQTLs, we performed an ABC-model based analysis utilising epigenomic features and Micro-C data to predict the enhancer-gene connections. Accepting these caveats, our analysis enabled us to nominate candidate gene targets as the biological basis of risk loci. Only *TERT* (albeit promoter mutations) is an established ccRCC driver gene (i.e. recurrent nonsynonymous somatic mutations under positive selection). This implies a model by which genetic predisposition indirectly affects oncogenesis and specifically after the acquisition of primary driver mutations in progenitors.

Collectively our observations and those of previously published researchers highlights central pathways as being central to mediating inherited risk, notably linkage between hypoxia sensing (*VHL-EPAS1-BHLHE41*) and cell cycle genes (*MYC-CCND1*) as well as PAX8 and HIF as mediating TFs. Additionally, we provide further evidence for the importance of genetic variation in telomerase maintenance in mediating ccRCC as well as additional support for DNA damage genes, *CHEK2* and *FANCD2*, in disease aetiology (Fig. 4).

In addition to emphasising the role of genetic variation in established ccRCC genes and pathways, we identify candidate target genes with hitherto no previously well-established role. These include *KCNQ1* which encodes a potassium channel, mutations of which have previously been linked to several cancers[71,72], including colorectal cancer[73]; apoptosis genes *GRAMD4* and *CASP9* which have well-established roles in a wide range of different cancers[74–76]; and *MYEOV*, over-expression of which has been associated with multiple cancers[36,37,77–79]. Although speculative, *INSR* plays a role in insulin signalling, and its linkage to ccRCC raises the possibility that it may in part mediate the obesity associated risk of RCC.

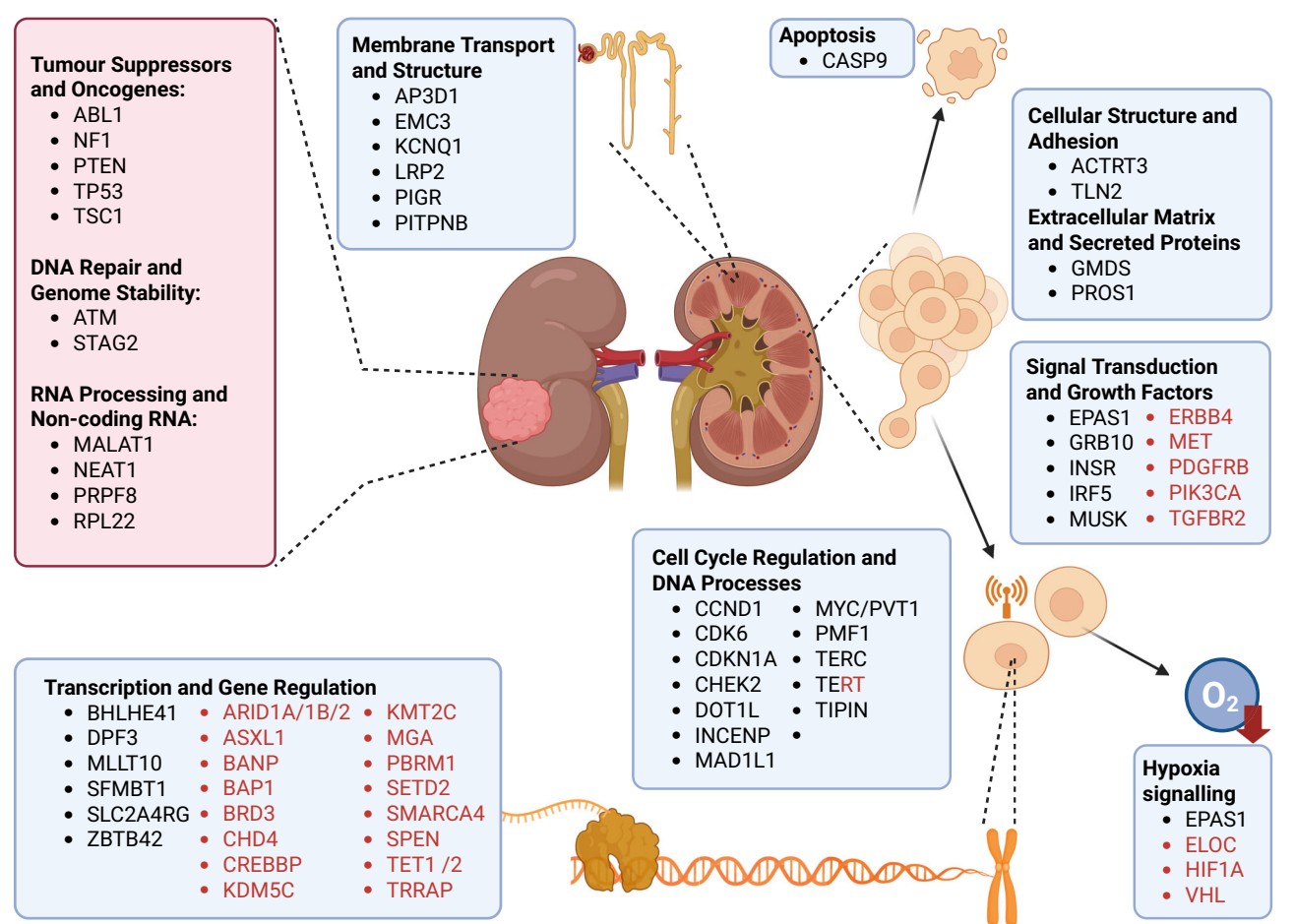

**Fig. 4 | Molecular functions of genes implicated in ccRCC risk.** The boxes indicate target genes implicated by GWAS and functional data. Included in red boxes and text are genes implicated in somatic studies. *TERT* has been implicated in both.

A key goal of cancer GWAS is to uncover genes and mechanisms that could serve as clinically actionable targets. Several pathways converge on MYC activation as a central feature of ccRCC predisposition, yet MYC remains a challenging therapeutic target. Hence the exploration of other mechanisms to indirectly target the MYC-axis would be entirely appropriate, and it is noteworthy that our study has identified several genes which could afford this opportunity. Telomere dysregulation is also critical to the development of ccRCC. Inhibitors of telomerase have been shown to reactivate telomere shortening and several approaches are being pursued therapeutically to achieve telomerase suppression, including targeting the RNA component of telomerase, chemical inhibition of telomerase, small molecule targeting of TERT, and telomerase vaccines. In some reports replicative senescence and apoptotic cell death of tumour cells has been achieved while having little or no effect on normal diploid cells.

In summary, our analysis expands our understanding of the genetics of ccRCC and we provide further insight into the functional basis of risk loci, implicating novel genes in the development of ccRCC, which raises the potential for therapeutic targeting.

## Methods
### Ethics
The GWAS summary statistics were from published work. Similarly, the eQTL data were obtained from publicly accessible resources, hence no additional ethical approval for the study was required.

### GWAS statistics and definition of risk loci
GWAS summary association statistics were obtained from the recently published meta-analysis GWAS study of 29,020 RCC cases and 835,670 controls[6]. Restricting our analysis to the clear cell renal carcinoma (ccRCC) subtype in European ancestry individuals provided data on up to 14,627 cases and 738,190 controls. Risk loci were defined by variants having a $P < 5 \times 10^{-8}$ that were at >500 kb apart. To identify secondary association signals, we performed a conditional analysis using Genome-wide Complex Trait Analysis with Conditional and Joint analysis (GCTA-COJO)[80] v1.93. In total, 52 autosomal variants were identified, which mapped to 46 independent risk loci. Gene distance and variant annotation information was obtained from HaploReg v4.1.

### Fine-mapping of risk loci
We performed statistical fine-mapping of these ccRCC risk loci using PolyFun[19] v1.4.1 and susieR[20] v0.11.92. In brief for each risk locus, we extracted variants within a 250 kb window and calculated the prior causal probabilities non-parametrically, using the established PolyFun protocol, which estimates the per-SNP heritability for each variant, weighted by their functional annotations. Annotation data was derived from the baseline-LF v2.2 annotation data provided by the A. Price group (https://alkesgroup.broadinstitute.org/LDSCORE/). LD scores were calculated using data from 4284 disease-free European individuals in the 1000 Genomes and UK10K Projects. Using the prior causal probabilities estimated by PolyFun we fine-mapped loci across a 500 kb window using the Sum of Single Effects (SuSiE) model, implemented in susieR. For loci with one independent variant, we set the maximum number of causal variants to two, as susieR is unable to use LD information for a single variant. For loci with multiple independent variants, we performed fine-mapping of the region including all independent variants and set the maximum number of causal variants equal to the number of independent variants. The output from susieR included a

Posterior Inclusion Probability (PIP) for each variant, and the 95% credible set that the variant belongs to. Variants with PIPs >0.001 and that cumulatively reached a probability of 0.95 were included in a credible set. To evaluate the deleteriousness of coding variants we used AlphaMissense[27] and CADD[81], where a CADD score ≥20 typically indicates a variant within the top 1% most deleterious in the human genome.

## Cell lines and cell culture

Renal cancer cell lines 786-O (CRL-1932, ATCC), A-498 (HTB-44, ATCC), UM-RC-2 (08090511, ECACC), and the immortalised normal renal epithelium cell line, HK-2 (CRL-2190, ATCC) were cultured in a humidified 5% $CO_2$ atmosphere under 20% $O_2$ (normoxic) or 1% $O_2$ (hypoxic) conditions balanced with $N_2$. 786-O cells were grown in Roswell Park Memorial Institute (RPMI) 1640 Medium (Gibco), A-498 cells in Minimum Essential Medium (MEM, Gibco), and UM-RC-2 cells in MEM medium supplemented with 1% non-essential amino acids (NEAA, Gibco). In all cases, media were also supplemented with 10% heat inactivated FBS (Sigma). HK-2 cells were grown in keratinocyte serum-free medium (KSFM, Gibco) supplemented with human recombinant epidermal growth factor (rEGF) and bovine pituitary extract (BPE) (Keratinocyte-SFM Supplement, Gibco). All cell lines were cultured to 90% confluency and passaged by using TrypLE (Gibco). We used whole genome sequencing to perform STR profiling to authenticate our cell lines and were routinely checked for Mycoplasma contamination (LookOut Mycoplasma PCR Detection Kit, Sigma-Aldrich).

## ChIPmentation

ChIPmentation was performed on histone marks H3K4me1 (C15410194, Diagenode, lot A1862D), H3K4me3 (C15410003, Diagenode, lot A8034D), H3K27ac (C15410196, Diagenode, lot A1723-0041D), H3K27me3 (C15410195, Diagenode, lot A0824D), H3K36me3 (C15410192, Diagenode, lot A1845P), CTCF (C15410210, Diagenode, lot A2354-00234P), and HIF1A (PA1-16601, Invitrogen, lot XB3514114) for the 786-O, A-498, UM-RC-2 and HK-2 cell lines, which were cultured under normoxic and hypoxic conditions, using a published protocol with minor modification[82]. In brief, $4 \times 10^7$ cells were fixed with 1% methanol-free formaldehyde for 10 min, quenched with 250 mM glycine for 5 min, washed with PBS and the cell pellet flash frozen in liquid nitrogen and stored at -80°C. Pellets were thawed and cells lysed twice in cytoplasmic buffer (50 mM HEPES, 150 mM NaCl, 1 mM EDTA, 1% Triton X-100, 0.1% sodium deoxycholate and 0.1% SDS) with protease inhibitors for 15 min at 4°C and centrifuged. Nucleic buffer (50 mM HEPES, 150 mM NaCl, 1 mM EDTA, 1% Triton X-100, 0.1% sodium deoxycholate and 1% SDS) was added to the pellet and incubated for 15 min at 4°C. Following centrifugation, the pellet was resuspended in 2 ml sonication buffer (10 mM Tris, 1 mM EDTA and 0.1% SDS) and transferred to Adaptive Focused Acoustics (AFA) militubes (Covaris). Chromatin was sonicated for 15 min using a Covaris E220, and dilution buffer (Tris 50 mM, NaCl 225 mM, NaDOC 0.15%, Triton X-100 1.5%) added to a 1:2 ratio. 5 µg of each antibody was added per $10^6$ cells and samples rotated overnight at 4°C. Protein G Dynabeads (ThermoFisher Scientific, USA) were washed in dilution buffer for 1 h prior to being added to samples (30 ml/sample), which were then rotated for 2 h at 4°C. Beads were washed once with WBI (50 mM HEPES, 150 mM NaCl, 1 mM EDTA, 1% Triton X-100, 0.1% sodium deoxycholate and 0.1% SDS), WBII (50 mM HEPES, 500 mM NaCl, 1 mM EDTA, 1% Triton X-100, 0.1% sodium deoxycholate and 0.1% SDS) and WBIII (10 mM HEPES, 250 mM NaCl, 1 mM EDTA, 0.5% sodium deoxycholate and 0.5% NP-40), and twice with TE and Tris-HCL (pH = 8) buffers. Washed bead-bound chromatin was resuspended in 1X Tagment DNA buffer with 1 µl Tagment DNA enzyme (Diagenode, Belgium) and incubated at 37°C for 10 min at 1400 rpm, before being placed on ice. Beads were then washed twice with WBI and resuspended in water before barcoded NGS libraries were generated using UDI for Tagmented Libraries (Set I-III; Diagenode, Belgium) and Kapa HiFi PCR master mix (Roche, Switzerland). For the input chromatin samples, 12 µl of the initial sonicated chromatin was incubated with the Tagment DNA enzyme at 37°C for 10 min, and sequencing libraries prepared as before. Libraries were cleaned, and size selected using Kapa Pure beads (Roche, Switzerland). The size and concentration of each library was analysed using an Agilent Bioanalyzer DNA High Sensitivity chip (Agilent Technologies, USA), and the libraries were pooled and quantified using the NEB Library Quant Kit (NEB) in conjunction with QuantStudio7 Real Time PCR (Thermo Fisher, USA). Each library was sequenced in NovaSeq 6000 (Illumina, USA) as 100 bp single end reads for ~50 million reads per library. Data processing was performed using the Nextflow nf-core chipseq pipeline[83] v1.2.1. The differential binding analysis was performed using the DiffBind package[84] v3.16.

Additional HIF and PAX8 binding data were obtained from Salama et al.[12] and Patel et al.[11], respectively. Peak region bed files were converted to GRCh38 coordinates using the UCSC liftOver tool.

## Omni ATAC

ATAC-sequencing was performed using a previously published protocol[85] on 786-O, A-498, UM-RC-2, and HK-2 cell lines cultured under both normoxic and hypoxic conditions. $5 \times 10^5$ cells were lysed in RSB containing 0.1% NP40, 0.1% Tween-20, and 0.01% digitonin, and incubated for 4 min on ice. Lysis was stopped by addition of 1 ml RSB buffer with 0.1% Tween-20 prior to centrifugation for 10 min. Pellets were incubated with Tagment DNA enzyme (Diagenode, Belgium) for 30 min at 37°C, while shaking at 1000 rpm. Tagmented DNA was cleaned using the Zymo DNA Clean & Concentrator kit (Zymo Research, US). Barcoded libraries were generated using UDI for Tagmented Libraries (Set I-III; Diagenode, Belgium) and Kapa HiFi PCR master mix (Roche, Switzerland). Libraries were cleaned using the Zymo DNA Clean & Concentrator kit (Zymo Research, US) and size selected by DNA agarose gel electrophoresis and Zymoclean Gel DNA Recovery Kit (Zymo Research, US). The size and concentration of each library was analysed using an Agilent Bioanalyzer DNA high sensitivity chip (Agilent Technologies, USA), afterwich libraries were pooled, and the final concentration was quantified using the NEBNext Library Quant kit (NEB) in conjunction with QuantStudio7 Real Time PCR (Thermo Fisher, USA). Each library was sequenced using a NovaSeq 6000 (Illumina, USA) as 50 bp paired end reads yielding ~50 million reads per library. Data processing was performed using the Nextflow nf-core atacseq pipeline[86] v1.2.1.

## RNA extraction and sequencing

RNA sequencing of 786-O, A-498, UM-RC-2, and HK-2 cell lines cultured under both normoxic and hypoxic conditions was performed. RNA was extracted from $10^6$ cells using the RNeasy Mini Kit (Qiagen) and quantified by Qubit 3.0 fluorometer (Life Technologies) and the 2100 Bioanalyser (Agilent Technologies), using the RNA Quantification Broad Range kit (Invitrogen) and the RNA 6000 Nano Reagents (Agilent), respectively. All samples had RNA integrity number >9.3. Samples were barcoded for library preparation with ribosomal RNA depletion and sequenced using an Illumina NovaSeq 6000 (Illumina, USA) as 100 bp paired-end for 50 million reads per library. Analysis of RNA-seq data was performed using the RNAflow pipeline[87] with default parameters.

## Micro-C

We generated Micro-C chromatin interaction maps of 786-O, A-498, UM-RC-2 and HK-2 cell lines under both normoxic and hypoxic conditions using a previously published protocol[88,89]. $1.2 \times 10^7$ cells were fixed in 3 mM DSG for 20 min at room temperature before addition of methanol-free formaldehyde for 10 min, followed by quenching of the reaction by addition of glycine for 5 min. Cell pellets were washed in 1x PBS, snap-frozen in liquid nitrogen and stored at -80°C before use. $10^6$ fixed cells were digested with micrococcal nuclease (MNase, Worthington), with incubation for 10 min at 37°C while shaking at 1000 rpm. The reaction was quenched by EGTA with incubation for 10 min at 65°C, shaking at 1000 rpm. To ensure adequate library complexity samples were pooled such that each replicate library had 2.5-3 × $10^6$ cells (total of 8 libraries). End repair and biotin dNTP labelling, proximity ligation, removal of unligated chromatin biotin ends and cross link reversal was performed as described previously. After overnight incubation at 65°C with Proteinase K to reverse crosslinking, DNA

was eluted from samples using phenol-chloroform and run on a 2.5% TAE gel. Bands of 250−400 bp size were excised and purified to ensure di-nucleosome DNA could be isolated, and biotinylated DNA capture was performed using C1 streptavidin beads (Invitrogen). End repair and adaptor ligation was conducted in the DNA bound beads using NEBnext Library Prep Kits (NEB) and NEB Unique Dual Index primers to produce final libraries. Size selection was performed using Kapa Pure beads (Roche). Each library was quantified using Agilent Bioanalyzer DNA High Sensitivity Chips and pooled together. The final pooled library was quantified by NEB Library Quant Kits and the quality was checked before sequencing, by a MiSeq Reagent kit v2 (Illumina, US). The data were processed using JuicerTools to enumerate valid interactions. We required valid interactions of >90% for assignment as *cis*-contacts, of which 60−70% had to be short range contacts. If the parameters were satisfactory, the pooled library was sequenced using a NovaSeq 6000 (Illumina, US) to a depth of >350 million reads per library, using 100 bp paired-end sequencing. FitHiC2[90] was used to call significant interactions. TADs and compartments were identified using hicFindTads at 10 kb resolution across the different cell lines[91].

### Cell-type specificity of risk variants

To identify the cell types through which RCC risk variants exert their effects, we analysed single-cell gene-expression profiles across different tissues using the Tabula Sapiens[16] v4 dataset (~500,000 cells from 24 organs from 15 normal human subjects). We used scDRS[17] v1.0.1 to link the scRNA-seq data with disease risk at a single-cell resolution, independent of cell type. In brief, using the GWAS association summary statistics, MAGMA[92] v1.10 defined a putative set of disease genes. Using the top 1000 putative genes, a disease score was calculated as a function of the GWAS z-scores and the scRNA-seq expression values. Cell-specific association *P*-values were calculated by comparing normalised disease scores to an empirical distribution of normalised scores across all control gene sets and all cells.

For histone mark enrichment, we adapted the variant set enrichment method of Cowper-Sal lari et al.[93]. Briefly, for each risk locus, we identified variants with strong LD (defined as $r^2 \geq 0.8$) to the lead variant and termed these the associated variant set (AVS). ChIP-seq data for H3K4me1, H3K4me3, and H3K27ac chromatin marks from up to 127 cell types were obtained from the NIH Roadmap Epigenomics Project data, in addition to the ChIP-seq data generated in-house for 786-O, A-498, UM-RC-2, and HK-2. For each mark, the overlap of the positions of variants in the AVS and the ChIP-seq peaks was determined to produce a mapping tally. A null distribution was generated by randomly selecting variants with the same LD characteristics as the risk-associated variants, and a null mapping tally calculated. This process was repeated 50,000 times, and approximate *P*-values were calculated as the proportion of permutations where the null mapping tally was greater or equal to the AVS mapping tally.

### Transcription factor binding

We used TOBIAS[94] v0.14.0 to predict transcription factor binding. Using the ATAC-seq data from 786-O, A-498, UM-RC-2, and HK-2 cell lines in conjunction with the JASPAR 2024 core non-redundant TF motif database[22], filtered to exclude non-human motifs, we predicted whether there were any potential transcription factors bound to open chromatin.

### ChromHMM

We used ChromHMM[95] v1.24 to predict chromatin states from H3K4me1, H3K4me3, H3K27ac, H3K27me3, and H3K36me3 histone marks. The BAM files from the nf-core chipseq pipeline were binarised and a 12-state model predicted for each cell line under both normoxic and hypoxic conditions. Chromatin states were annotated as per previously published work[95–97].

### Activity-By-Contact modelled prediction of enhancer-gene interactions

To predict enhancer-gene interactions, we used ABC[15] v1.1.0 in conjunction with the ATAC-seq, H3K27ac ChIP-seq, Micro-C, and RNA-seq data for each cell line. The analysis was performed as previously described using default parameters. Briefly, we examined the 150,000 enhancer-gene interactions having the highest ABC score for all enhancer regions within 2 Mb of the transcription start site (TSS) of a gene. We additionally obtained ABC predictions from ENCODE based on 811 adult tissues and cell lines[35]. For each sample, we identified the predicted enhancer regions that overlapped with the RCC enhancers, and additionally had the same target gene.

### Summary-data-based Mendelian Randomization (SMR)

To identify causal relationships between gene expression levels and the risk of RCC, we performed SMR[31] v1.3.1 analysis using eQTL results from GTEx Kidney Cortex (data derived from 73 healthy individuals)[32], the Susztak lab Kidney Biobank (data derived from a meta-analysis of 686 non-neoplastic kidney tissues)[33], and kidney renal cell carcinoma (KIRC) from TCGA (527 individuals)[34]. To select statistically significant associations between variants and genes, results within 100 kb of the risk variants with $P_{SMR} < 0.05$ and $P_{HEIDI} \geq 0.05$ were retained.

Region plots were created using pyGenomeTracks[98]. For Micro-C plots where cooler format data were required, we used the hicConvertFormat[91] tool to convert the hic matrices.

### Target gene prioritisation

We prioritised the gene mapping evidence in the following order (highest priority first): coding variants, SMR, ABC, and finally the closest gene. SMR served as a validation of eQTLs from patient data, whereas ABC predictions leverages renal-specific epigenomic and Micro-C data for high-resolution enhancer–gene assignments. When multiple gene predictions were available, the consensus prediction was selected. Where discordant predictions occurred, the union of all putative genes was retained. If ABC was the only prediction source, the highest scoring prediction was used. For the remaining loci, we assigned the closest gene as lower confidence gene predictions.

### Association with leucocyte telomere length

Two-sample Mendelian Randomisation (2S-MR) was used to examine the relationship between leucocyte telomere length (LTL; exposure) and ccRCC risk (outcome) using the TwoSampleMR package[48,49] v0.6.9. Association data for LTL were obtained from Codd et al.[47], and consisted of data on 472,174 individuals. Variants were considered potential instruments if they were associated at $P < 5 \times 10^{-8}$, with a minor allele frequency > 0.01. To avoid multicollinearity, correlated variants ($r^2 \geq 0.01$) were excluded. For each variant, effect estimates, and standard errors were retrieved, and were harmonised with the ccRCC outcome data, excluding any variants that were palindromic or had an ambiguous minor allele frequency (between 0.4-0.6). This resulted in 145 variants used in the MR analysis. For each variant, causal effect estimates were generated as odds ratios per one standard deviation unit increase in LTL ($OR_{SD}$), with 95% confidence intervals (CI), using the Wald ratio. Causal effects were also estimated using a random-effects inverse weighted variance (IVW-RE) model, which assumes each variant identifies a different causal effect. To assess robustness, we additionally compared causal estimates and associated *P*-values using weighted median (WME) and weighted mode-based (WMBE) methods. MR-Egger regression showed there was no directional pleiotropy (Egger intercept = -0.0060, standard error = 0.0043, $P_{Egger} = 0.16$), and the Steiger test showed the direction of causal effect for exposures was correct ($P_{Steiger} = 0$). We estimated the proportion of variance explained (PVE) using the Cancer Research UK lifetime risk estimate for RCC (2%). A leave-one-out strategy under the IVW-RE model was employed to assess the potential impact of outlying and pleiotropic variants, with no single variant impacting the analysis. We calculated power across a range of odds ratios, and showed it was sufficiently powered for OR > 1.25.

To test if pleiotropic associations reflect shared variants, we performed a colocalisation analysis using the coloc package[50,51] v5.2.3 across a 500 kb region around the lead variants. Coloc enumerates four possible configurations of causal variants for two traits, calculating support for each model

based on a Bayes factor. Adopting default prior probabilities, a posterior probability ≥0.80 was considered as supporting a specific model. We performed a genetic correlation between RCC and LTL using LDAK[52] v5.2, using the sum-her function with a prevalence of 0.02 (lifetime risk) and an ascertainment of 0.01 (ratio of cases to controls).

## Polygenic risk scores

We calculated a polygenic risk score (PRS) for ccRCC using the GWAS data and LDpred2. We generated PRS for 778 patients enroled in the Genomics England 100,000 Genomes Project with primary ccRCC[5]. All calculations were based on 1,444,196 HapMap3+ variants using bigsnpr[99] v1.12.18, specifically using the auto model, and the LD reference was based on European individuals in the UK Biobank. Individuals were categorised as "highPRS" or "lowPRS" if they had a PRS in the top or bottom two quintiles of PRS values, respectively. Association with these categories were associated with clinical features using logistic regression with age at sampling, laterality, stage, tumour grade, size and ploidy, telomere length (determined using Telomerecat[100] and TelomereHunter[101]), and single base substitution (SBS) and double-base-substitution (DBS) and indel (ID) signatures (IDS) mutational signatures as covariates. Details of these analyses have been previously described[5].

We additionally interrogated these candidate genes to evaluate the impact of mutations on patient outcome. For the genes that were mutated in >10 patients, we performed a univariate linear regression for grade and stage using the statsmodels Python package. After confirming the model assumptions of a Cox proportional-hazards test (i.e. the hazard ratio between individuals remains constant over time), and adjusting for age and sex, we assessed if there was an association between mutation status and survival, using the lifelines Python package.

In all these analyses a two-sided *P*-value < 0.05 was considered as being statistically significant.

## Gene evidence

We used oncoEnrichR[59] v1.4.2 to analyse the gene sets. This tool provides a suite of analyses, incorporating data from several resources, including cancer associations, drug associations, synthetic lethality, gene fitness, and protein-protein interactions.

Regulatory interaction data were obtained from the OmniPath[102]/ DoRothEA resource. This dataset contains a list of previously identified TF-target interactions that are scored based on multiple lines of evidence (namely literature-curated resources, ChIP-seq peaks, TF binding site motifs, and gene expression-inferred interactions). Regulatory interactions were inferred using gene expression in TCGA tumour samples.

Protein-protein interactions were determined using the STRING database[44] v12.0. Cell viability and gene essentiality data were obtained from The Cancer Dependency Map (DepMap, 2020_Q2 release) which provides information on a systematic genome-scale CRISPR/Cas9 drop-out screen in 912 cancer cell lines. Drug tractability information was based on data from the Open Targets Platform, and pathway enrichment was performed using clusterProfiler with respect to Gene Ontology terms, KEGG, Reactome, and the Molecular Signatures Database (MSiGDB).

## Statistics and reproducibility

All statistical analyses performed were using publicly available open-source software with default parameters, and are described in the relevant sections. Data for generating Fig. 3, and Supplementary Figs. 1A, 1B, and 2 are provided in Supplementary Data 14.

## Reporting summary

Further information on research design is available in the Nature Portfolio Reporting Summary linked to this article.

## Data availability

Data generated in this study have been deposited at the European Genome-phenome Archive (EGA), which is hosted by the EBI and the CRG, under accessions EGAD50000001883 (RNA-seq), EGAD50000001884 (Micro-C), EGAD50000001885 (ATAC-seq), and EGAD50000001886 (ChIP-seq). GWAS data are available from the GWAS Catalogue (GCST90320058, GCST90320061, GCST90320064, GCST90320065). Single cell RNA-seq data were obtained from the Tabula Sapiens project (https://tabula-sapiens-portal.ds.czbiohub.org). HIF and PAX8 binding data were obtained from GEO (accessions GSE67237 and GSE163487, respectively). Transcription factor binding was based on data from JASPAR 2024 (https://jaspar.genereg.net). Functional annotations for the fine-mapping were provided by the A. Price group (https://alkesgroup.broadinstitute.org/LDSCORE). Histone marks in different tissues were obtained from the NIH Roadmap Epigenomics Project (https://egg2.wustl.edu/roadmap/web_portal). eQTL data were obtained from PancanQTL (http://bioinfo.life.hust.edu.cn/PancanQTL), GTEx (https://gtexportal.org), and the Susztak Lab Human Kidney eQTL Atlas (https://susztaklab.com/Kidney_eQTL/). Gene annotation data were obtained from OmniPath (https://omnipathdb.org), DoRothEA (https://saezlab.github.io/dorothea), DepMap (https://depmap.org) and Open Targets (https://www.opentargets.org), as implemented in oncoEnrichR (https://oncotools.elixir.no). Details regarding access to the Genomics England ccRCC patient data are provided in Culliford et al.[5].

## Code availability

No custom code was generated. Publicly available code was used for all aspects of data processing and analysis.

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

## Acknowledgements

R.S.H. acknowledges grant support from Cancer Research UK (C1298/A8362) and the Wellcome Trust (214388). This research was made possible through access to data in the National Genomic Research Library, which is managed by Genomics England Limited (a wholly owned company of the Department of Health and Social Care). The National Genomic Research Library holds data provided by patients and collected by the NHS as part of their care and data collected as part of their participation in research. The National Genomic Research Library is funded by the National Institute for Health Research and NHS England. The Wellcome Trust, Cancer Research UK and the Medical Research Council have also funded research infrastructure. Figure 4 was created using BioRender.

## Author contributions

M.M. and J.V. performed the laboratory experiments and analysed the data. P.J.L., C.M., and M.W. performed additional bioinformatics analyses. M.M., P.J.L., and R.S.H. wrote the manuscript. R.S.H. supervised the study. All authors read and approved the final version.

## Competing interests

The authors declare no competing interests.
