## [Transparent Peer Review file · Communications Biology]

Deciphering Genetic Susceptibility to Clear Cell Renal Cell Carcinoma

Corresponding Author: Professor Richard Houlston

Version 0:

Reviewer comments:

Reviewer #1

(Remarks to the Author)

Mandelia et al. present an post-GWAS analysis of Renal Cell Carcinoma risk loci. They use functional genomics and eQTL data to link risk variants to candidate target genes, highlighting enrichment pathways and potential therapeutic opportunities.

The background clearly presents the context and the research goals, the analytical choices are justified, and the interpretation of results is sound. Strengths include incorporation of multiple functional genomics data from trait-relevant models, including high-resolution chromatin conformation, into a modified ABC target gene prediction model. The list of predicted risk genes, which includes novel targets, is then taken into several data-mining analyses. Overall, this is a neat study, a good example of GWAS follow-up, and provides the basis for future functional evaluation studies.

Some minor points and questions I believe may strengthen the paper:

- The axis title $-\log_{10}$ (P value) in Supp figure 1A should be labeled as the adjusted P value, as per the text.
- Just to follow the TFBS analysis - HIF motifs are observed at 2 GWAS loci - do these motifs directly overlap risk SNPs? Is the motif prediction altered?
- Then, further on, HIF2A sites overlap was observed at 7p12, 8q24.21, 11q13.3 and 12p12.1. It might be nice to show the effect of the alleles on the TF PWMs, if there is direct overlap.
- What are the predicted effects of the missense variants in CCDC99 and INCENP? And are these the only candidate variants in these signals?
- How do the ABC predictions made with custom data compared to the published enhancer-gene pairs? Can any comment be made about cell type-specificity of targets? It's interesting to compare the nomination of targets based on the baseline (distance), so can a comment be made about this, and perhaps columns indicating the nearest gene and distance to the risk SNP added to ST4.
- For the analysis of potential druggable targets - are there any hints that the direction of gene expression or protein levels (e.g. from eQTL, MPRA) is amenable to the action of inhibitors or agonists?

Reviewer #2

(Remarks to the Author)

This study focuses on the risk loci identified through genome-wide association studies (GWAS) in clear cell renal cell carcinoma (ccRCC), and comprehensively characterizes the disease-relevant target genes through an integrative analysis of chromatin architecture (Micro-C), epigenomic profiles (ATAC-seq, ChIP-seq), and gene expression data (RNA-seq, eQTL, SMR). It represents a meaningful contribution to the field.

By applying multi-layered and cutting-edge methodologies to elucidate the molecular basis of genetic susceptibility in ccRCC, this work advances our understanding of renal cancer biology and holds potential for future therapeutic target discovery. The study presents well-organized data integration and is overall a solid and carefully conducted work. I would like to offer the following comments, which may be helpful in deepening the interpretation and discussion.

Major comments

1. This study focuses on a reanalysis of ccRCC GWAS data in individuals of European ancestry. Given that the effect sizes and allele frequencies of risk variants can vary across populations, it may be important to assess whether similar findings are observed—or whether markedly different associations emerge—in other ancestral groups (e.g., Asian or African populations). Such evaluation could provide valuable insights into whether the underlying tumorigenic mechanisms are universal or population-specific. Would it be possible to examine these loci in other ancestral populations as well?
2. The identification of missense variants in *CCDC99* and *INCENP* as candidate target genes is intriguing, as these genes are involved in cell division and may point to novel pathogenic mechanisms underlying ccRCC susceptibility. However, there appear to be limited reports on their dysregulation or functional relevance in ccRCC. Have the authors examined expression levels or somatic alterations of these genes using publicly available datasets such as TCGA? Also, do the authors consider functional validation of these variants to assess their potential role in tumorigenesis?
3. It is intriguing that enhancer–gene connections at risk loci were observed not only in RCC cell lines but also in normal renal epithelial cells, suggesting a shared chromatin architecture in renal cells. In such cases, how do the authors interpret the role of risk variants in contributing to tumorigenesis, given that these structural connections are already present in normal cells?
4. The use of multiple complementary approaches—such as the ABC model, eQTL analysis, SMR, and identification of missense variants—for target gene prioritization at each ccRCC risk locus is compelling. However, it was not entirely clear to what extent the results from these different methods were concordant. In cases where they diverged, how did the authors prioritize one set of findings over another? Could the authors elaborate on how they assessed the reliability of each approach, and how such assessments guided the final selection of target genes?
5. Some of the identified target genes, such as *CHEK2*, were found to be downregulated in tumors compared to normal kidney tissue, which is particularly intriguing. These findings suggest a potential role in tumorigenesis. Would it be possible to examine whether alterations in these newly identified target genes are associated with clinical features such as tumor grade, stage, or patient survival? Such an analysis could further strengthen the clinical relevance of the findings.

Minor comments

6. Line 107 uses "EPAS", whereas all other instances refer to "EPAS1". For consistency, it may be better to unify the notation as "EPAS1".
7. In Figure 4, similar to the above, "EPAS" is listed under "Signal transduction and growth factors", while "EPAS1" is listed under "Hypoxia signaling". As the inconsistent notation may cause confusion, we recommend unifying the labels using the official gene name "EPAS1".
8. In Supplementary Figure 1A, two kidney epithelial cell types are shown: "kidney epithelial cell (epithelial)" and "kidney epithelial cell (kidney)", but the distinction between them is not entirely clear. Since only one of these cell types shows a significant association, it may help prevent confusion if the difference is clarified in the legend or main text.
9. In Supplementary Figure 3, the gene names appear quite small and difficult to read. It may be helpful to enlarge the text for better visibility.

Version 1:

Reviewer comments:

Reviewer #1

(Remarks to the Author)

Thank you to the authors for the changes to the manuscript. I have no further questions or concerns.

Reviewer #2

(Remarks to the Author)

The authors have addressed the reviewers' comments appropriately and thoroughly. I have no further comments. I would like to express my respect for the authors' achievements.

Reviewer #1 (Remarks to the Author):

Some minor points and questions I believe may strengthen the paper:

- The axis title '-log₁₀ (P value)' in Supp figure 1A should be labeled as the adjusted P value, as per the text.

Response: This typographical error has been corrected to reflect the adjusted P-value in Supplementary Figure 1A, ensuring consistency with the text.

- Just to follow the TFBS analysis - HIF motifs are observed at 2 GWAS loci - do these motifs directly overlap risk SNPs? Is the motif prediction altered?

- Then, further on, HIF2A sites overlap was observed at 7p12, 8q24.21, 11q13.3 and 12p12.1. It might be nice to show the effect of the alleles on the TF PWMs, if there is direct overlap.

Response: We used motifbreakR (Coetzee *et al.*, 2015; Bioinformatics; doi:10.1093/bioinformatics/btv470) to determine if there was a functional impact of risk variants on transcription factor binding sites. For the two GWAS loci (12p12.1 and 22q13.31) with predicted HIF1A motifs, only the variant at the 22q13.31 locus (rs714024) was predicted to show significant allele-specific binding affinity (P-value = 6.81×10^{-5}). Investigating the HIF ChIP-seq data, the 22q13.31 locus additionally was bound by ARNT, and was predicted to disrupt the binding of this motif (P = 6.7×10^{-5}), along with rs6442146 (3p25.3, P = 4.6×10^{-5}). Additionally, at loci overlapping HIF1A ChIP peaks, rs6442146 (3p25.3, P = 1.0×10^{-3}) and rs11643164 (16q12.1, P = 3.6×10^{-5}) were predicted to disrupt binding. No EPAS1 ChIP-seq peak overlaps with risk variants showing significant allele-specific effects on EPAS1 binding. These results highlight potential allele-specific modulation of HIF binding, potentially altering transcriptional regulation in ccRCC. The text has been revised to include this information.

- What are the predicted effects of the missense variants in CCDC99 and INCENP? And are these the only candidate variants in these signals?

Response: As requested, we now provide information on the predicted effects on these missense variants. Specifically, in the revised text we now state "The missense variants rs2277283 (M>T, INCENP, 11q12.3) and rs116483731 (R>Q, SPDL1 (also known as CCDC99), 5q35.1) were identified as likely causal. CADD predicted rs2277283 as pathogenic (CADD score: 23.5), supported by AlphaMissense. As INCENP is an inner centromere protein, this change suggests potential disruption of the metaphase-anaphase transition. rs116483731 (CADD score: 24.2) was predicted to be deleterious by CADD but benign by AlphaMissense, yet its role in mitotic spindle formation and chromosome segregation, combined with rare INCENP germline mutations previously being linked to juvenile nephronophthisis, underscore their relevance to renal biology."

- How do the ABC predictions made with custom data compared to the published enhancer-gene pairs? Can any comment be made about cell type-specificity of

targets? It's interesting to compare the nomination of targets based on the baseline (distance), so can a comment be made about this, and perhaps columns indicating the nearest gene and distance to the risk SNP added to ST4.

Response: Regarding cell specificity, we compared the enhancer-gene predictions using our custom data with the predictions on 811 tissues or cell lines from ENCODE. While we did not observe substantial tissue specificity, there was variant specificity. For example, at the 16q12.1 locus, the enhancer region association with *HEATR3* was found across all cell types. In contrast, the 8q24.21 locus association with *PVT1*, *CASC11*, and *MYC* was only found in RCC and renal tissues. We present these data in Supplementary Figure 2.

In the case of several loci (35/52), the risk variant is intronic, and in these cases, there was often no strong ABC candidate. The closest gene was included in Supplementary Table 1, but we now include this information in a revised Supplementary Table 4.

- For the analysis of potential druggable targets - are there any hints that the direction of gene expression or protein levels (e.g. from eQTL, MPRA) is amenable to the action of inhibitors or agonists?

Response: We analysed the putative drug target genes in the eQTL datasets, and found that the risk allele of rs140527149 (1p36.21) is associated with reduced *CASP9* expression. Similarly, MPRA data indicates that rs4389139 ($r^2 = 0.88$ to candidate variant, rs11643164, 16q12.1) affects *HEATR3* expression, and rs2860183 ($r^2 = 0.99$ to candidate variant, rs11085214, 19p13.2) affects *INSR* expression in RCC. The MPRA association supported a relationship between rs77247065 and *CCND1* expression but was formally not significant after correction for multiple testing ($P_{adj} = 0.08$). Additionally in the RNA-seq data, 19 of the 29 potentially druggable genes, including *CDKN1A*, *TERT*, *CCND1*, and *INSR* were differentially expressed in RCC compared to normal kidney cells. These data indicate the suitability of many of these genes as inhibitors or agonists, and this information is now included in Supplementary Table 12.

Reviewer #2 (Remarks to the Author):

Major comments

1. This study focuses on a reanalysis of ccRCC GWAS data in individuals of European ancestry. Given that the effect sizes and allele frequencies of risk variants can vary across populations, it may be important to assess whether similar findings are observed—or whether markedly different associations emerge—in other ancestral groups (e.g., Asian or African populations). Such evaluation could provide valuable insights into whether the underlying tumorigenic mechanisms are universal or population-specific. Would it be possible to examine these loci in other ancestral populations as well?

Response: We focused on European ancestry GWAS data due to the larger sample size (14,627 cases, 738,190 controls), and hence greater statistical

power for fine-mapping and target gene identification. Additionally, this was the ancestry of the majority of the available annotation data. The original GWAS meta-analysis by Purdue et al. (2024) included non-European populations (namely, Asian, Latin American, and African ancestries). We performed a preliminary analysis of the 52 identified loci using summary statistics from Asian (621 cases, 86,796 controls), Latin American (1,277 cases, 2,180 controls), and African (417 cases, 3,109 controls) ancestry groups from Purdue et al. (2024). Of the 52 loci, 6 showed nominal significance ($P < 0.05$) in Asian populations, 14 in Latin American populations, and 7 in African populations, with effect sizes generally consistent but weaker due to lower power. Notably, rs10908176 (CCND1, 11q13.3) and rs11085214 (INSR, 19p13.2) showed associations across multiple ancestries, suggesting shared tumorigenic mechanisms for cell cycle and regulation. However, loci like rs116483731 (SPDL1, 5q35.1) and rs143459581 (TTC28, 22q12.1) were specific to European ancestry, potentially reflecting population-specific regulatory variants. Conversely, as previously observed, rs7629500 (3p25.3) is predominantly found in African ancestry populations (RAF = 0.1, <0.001 in European ancestry populations) and localises within the 3'-UTR of VHL, a known driver of ccRCC development. These results are now summarised in a new Supplementary Table 13, acknowledging that larger non-European cohorts are needed for comprehensive validation.

2. The identification of missense variants in CCDC99 and INCENP as candidate target genes is intriguing, as these genes are involved in cell division and may point to novel pathogenic mechanisms underlying ccRCC susceptibility. However, there appear to be limited reports on their dysregulation or functional relevance in ccRCC. Have the authors examined expression levels or somatic alterations of these genes using publicly available datasets such as TCGA? Also, do the authors consider functional validation of these variants to assess their potential role in tumorigenesis?

Response: Based on TCGA data (using the Pan-Cancer Atlas), *INCENP* is mutated in 1% of samples (5/510), with 1 sample being homozygously deleted. *SPDL1* (*CCDC99*) is mutated in only 2 samples. These small numbers prevent any confident conclusions relating to the effect on expression by the mutations. Given the rarity of these mutations, we believe that any functional validation is beyond the scope of our present study.

3. It is intriguing that enhancer–gene connections at risk loci were observed not only in RCC cell lines but also in normal renal epithelial cells, suggesting a shared chromatin architecture in renal cells. In such cases, how do the authors interpret the role of risk variants in contributing to tumorigenesis, given that these structural connections are already present in normal cells?

Response: The shared enhancer-gene connections shown in RCC (786-O, A-498, UM-RC-2) and normal renal (HK-2) cell lines suggest that risk variants modulate pre-existing regulatory architecture rather than acting as primary oncogenic drivers per se. These variants are therefore likely to fine-tune gene expression (e.g. via altered TF binding or enhancer activity) in pathways such as hypoxia sensing (EPAS1) or cell cycle (CCND1), predisposing cells to tumourigenesis upon additional somatic alterations (e.g. VHL loss). For instance, at 11q13.3, the risk variant enhances *CCND1* expression via EPAS1 binding, which is present in both normal and tumour cells but amplified in RCC due to hypoxic conditions. This model of predisposition is supported by the absence of recurrent somatic driver mutations in most of the target genes (except *TERT*), indicating their role in early susceptibility rather than direct transformation.

4. The use of multiple complementary approaches—such as the ABC model, eQTL analysis, SMR, and identification of missense variants—for target gene prioritization at each ccRCC risk locus is compelling. However, it was not entirely clear to what extent the results from these different methods were concordant. In cases where they diverged, how did the authors prioritize one set of findings over another? Could the

authors elaborate on how they assessed the reliability of each approach, and how such assessments guided the final selection of target genes?

Response: We prioritised the gene mapping evidence in the following order: coding variants, SMR, ABC, and finally the closest gene. These categories were generally mutually exclusive. For instance, missense variants generally lacked SMR associations or high-confidence ABC predictions. Similarly, several genomic loci with an SMR result did not demonstrate a confident enhancer-gene link. For example, the 4q25 region was exclusively associated with ETNPPL by SMR. When multiple gene predictions were available, the consensus prediction was selected. For example, at the 22q13.31 locus, where SMR indicated GRAMD4 and ABC implicated both CERK and GRAMD4, GRAMD4 was chosen. Where discordant predictions occurred, the union of all putative genes was retained. Based on these data combined with the genes containing coding variants we identified high confidence target genes for 28 of the 52 ccRCC risk loci. For the remaining loci, we assigned the closest gene as lower confidence gene predictions, 16 loci of which fell within introns of genes. As baseline, 37 genes were the closest gene.

5. Some of the identified target genes, such as CHEK2, were found to be downregulated in tumors compared to normal kidney tissue, which is particularly intriguing. These findings suggest a potential role in tumorigenesis. Would it be possible to examine whether alterations in these newly identified target genes are associated with clinical features such as tumor grade, stage, or patient survival? Such an analysis could further strengthen the clinical relevance of the findings.

Response: We investigated this in the Genomics England ccRCC dataset consisting of 778 patients with primary tumours. Of the genes identified in this analysis, only 7 genes had pathogenic or likely pathogenic germline mutations: TERT (12%), CHEK2 (3%), MAD1L1 (2%), INSR (1%), PROS1 (0.4%), LRP2 (0.3%), KCNQ1 (0.1%). Of those that had >10 samples with mutation (TERT, CHEK2, MAD1L1, INSR), we performed univariate linear regression for grade and stage, and none showed a significant association. Similarly, we performed a Cox proportional-hazards model test (with all passing the model assumptions), adjusting for age and sex, and mutation status in none of the genes were significantly associated with overall patient survival.

Minor comments

6. Line 107 uses "EPAS", whereas all other instances refer to "EPAS1". For consistency, it may be better to unify the notation as "EPAS1".

Response: We have edited the text to standardise naming of genes. We have also standardised the gene names to the approved HGNC nomenclature (e.g. EPAS1 instead of HIF2A).

7. In Figure 4, similar to the above, "EPAS" is listed under "Signal transduction and growth factors", while "EPAS1" is listed under "Hypoxia signaling". As the inconsistent

notation may cause confusion, we recommend unifying the labels using the official gene name "EPAS1".

Response: We have corrected the figure to use the official name, EPAS1.

8. In Supplementary Figure 1A, two kidney epithelial cell types are shown: "kidney epithelial cell (epithelial)" and "kidney epithelial cell (kidney)", but the distinction between them is not entirely clear. Since only one of these cell types shows a significant association, it may help prevent confusion if the difference is clarified in the legend or main text.

Response: The "kidney epithelial cell (epithelial)" and "kidney epithelial cell (kidney)" correspond to the same set of cells. scDRS calculates p-values empirically by comparing the normalised disease score in the given cells compared to all other cells in the analysis. As the analysis was performed for each tissue independently, these kidney epithelial cells were compared both to all other kidney cells ("kidney epithelial cell (kidney)") as well as all other epithelial cells ("kidney epithelial cell (epithelial)"). When the figure was generated, it pulled all results that contained "kidney". We have removed the "kidney epithelial cell (epithelial)" to avoid confusion.

9. In Supplementary Figure 3, the gene names appear quite small and difficult to read. It may be helpful to enlarge the text for better visibility.

Response: We have increased the font size of the labels.